# Trachoma prevention practice and associated factors among mothers having children aged under nine years in Andabet district, northwest Ethiopia, 2022: A multi-level analysis

Zufan Alamrie Asmare[1]*, Natnael Lakachew Assefa[2], Dagmawi Abebe[3], Solomon Gedlu Nigatu[4], Yezinash Addis Alimaw[2]

1 Department of Ophthalmology, School of Medicine and Health Science, Debre Tabor University, Debre Tabor, Ethiopia, 2 Department of Optometry, School of Medicine and Health Science, University of Gondar, Gondar, Ethiopia, 3 Department of Ophthalmology, School of Medicine and Health Science, University of Gondar, Gondar, Ethiopia, 4 Department of Epidemiology & Biostatistics, Institute of Public Health, College of Medicine and Health Science, University of Gondar, Gondar, Ethiopia

* zufanalamrie2@gmail.com

**Data Availability Statement:** All relevant data are within the paper and its Supporting Information files.

## Abstract

### Background

The world health organization (WHO) adopted the Surgery, Antibiotic, facial cleanliness, and environmental improvement (SAFE) strategy for the prevention of trachoma, and different prevention strategies have been employed in Andabet district. Trachoma still has a high prevalence despite these efforts. So, it is imperative to assess ground trachoma prevention practice (TPP) since there are insufficient studies in the study area.

### Objective

To determine the magnitude and factors associated with TPP among mothers having children aged under nine years in Andabet district, Northwest Ethiopia.

### Method

A community-based cross-sectional study involving 624 participants was conducted June 1–30, 2022. Systematic random sampling was carried out to select study participants. Multi-level binary logistic regression analysis was used to identify factors associated with poor TPP. Descriptive and summary statistics were performed and variables with p-value < 0.05 in the best-fitted model were declared to be significantly associated with poor TPP.

### Results

In this study, the proportion of poor TPP was found to be 50.16% (95%CI = 46.23, 54.08). In the multi-variable multi-level logistic regression; having no formal education (AOR = 2.95; 95%CI: 1.41,6.15) and primary education (AOR = 2.33; 95%CI:1.04, 5.24), being a farmer

**Funding:** The University of Gondar funded this study (grant number: 556/2022, grant recipient: ZAA). The funders had no role in study design, data collection and analysis, decision to publish, or preparation of the manuscript.

**Competing interests:** The authors have declared that no competing interests exist.

(AOR = 3.02; 95%CI:1.73,5.28), and merchant (AOR = 2.63; 95%CI:1.20, 5.75), time taken to water point >30 minutes (AOR = 4.60,95CI:1.30,16.26) and didn't receive health education about trachoma (AOR = 2.36;95CI:1.16,4.79) were significantly associated with poor TPP.

## Conclusion

The proportion of poor TPP was high relative to other studies. Level of education, occupation, time taken to the water point, and health education were significantly associated with poor TPP. Therefore, taking special attention to these high-risk groups could decrease the poor TPP.

## Author summary

Trachoma prevention and control strategies have been successful in certain societies, but it has been more difficult in many communities. In Ethiopia, there are 10.2 million cases of trachoma, in which the Amhara region takes the lion's share. Half the global population requiring intervention for trachoma elimination is in the country where some regions/districts have up to 37%TF rate after years of antibiotic treatment. Facial cleanliness & Environmental improvements are critical for sustained progress toward elimination. This study demonstrates the need to consider support for the introduction of those interventions (F and E) for trachoma elimination in Ethiopia and thus the elimination of an estimated half the global burden. The findings can be used to establish effective public health approaches and implementation of those strategies (F and E) for trachoma prevention and control.

## Introduction

Trachoma is caused by *Chlamydia trachomatis*, a neglected tropical disease [1]. It has been one of the most debilitating diseases affecting 60 to 90% of children, especially those under nine years [2,3]. As a devastating disease, Trachoma hinders the performance of children in school and impairs their ability to lead a healthy and productive life [2]. Moreover, these ruinous diseases can result in vision loss, stigma, reduced productivity, and economic loss of US$ 2.9–5.3 billion annually, and half of the global burden of trachoma is shared by Ethiopia solely [4,5].

Globally, trachoma causes 1.9 million visual impairments and 1.2 million blindness. It is still endemic in 44 countries [6,7]. More than 80% of active trachoma is concentrated in Africa, nearly half is in sub-Saharan Africa [4]. In Ethiopia, there are 10.2 million cases of trachoma, in which the Amhara region takes the lion's share (62.6%) [4,8].

To control the disease, WHO implemented SAFE (surgery for advanced disease, Antibiotics, facial cleanliness, and Environmental improvement) [9–12]. As a result of the implementation, trachoma prevalence has drastically decreased [4,9]. As of 5 October 2022, fifteen countries have been validated as having eliminated the disease as a public health problem [13]. Aside from S (surgery) and A (antibiotics), Proper practice of F (facial cleanliness) and E (Environmental improvement) is responsible for 58.7% and 37.4% reduction of trachoma prevalence at all ages and in children respectively [14]. Therefore, changes in hygiene behavior

and improvements in environmental infrastructure are ideal long-term strategies for trachoma control [15].

Despite these, in our study area Andabet, Northwest Ethiopia, After 8 to 11 years of implementation of SAFE, the prevalence of TF (trachomatous follicular) was 37% in 2017, and it remained hyperendemic [8,16].

According to different studies conducted in Ethiopia, the prevalence of poor TPP was high, which ranges between 45.5 and 64.4% [1,17]. Shreds of evidence have shown that individual-level factors such as the age of the mother, husband's education, basic knowledge about trachoma, mother's attitude towards trachoma, taking health education about trachoma, time taken to the water point and frequency of getting water and also community-level factors such as residence and types of water source were affected TPP [1,7,16–21].

Although numerous studies were done on TPP, most of them did not consider the community-level factors that could affect TPP. Therefore, we aimed to determine the magnitude and associated factors of TPP. Identifying various factors at both individual and community levels can have a key role in implementing policies and programs aimed at minimizing poor TPP.

## Methods

### Ethics statement

The study adhered to the tenets of the Declaration of Helsinki and approval was sought and obtained from the Ethical Review Board of the College of Medicine and Health Sciences, University of Gondar (Reference Number: 1556/2022). A permission letter was obtained from Andabet district administrative office and written informed consent was obtained from all voluntary participants. The participants were informed that the study would not impose harm on them. There were no personal identifiers and the confidentiality of the study participants was maintained at all stages of data processing. Informed verbal consent was obtained from each respondent and confidentiality was kept by using codes and avoiding personal identifiers.

### Study area and period

This study was conducted in the Andabet district, south Gondar zone, Amhara region, Ethiopia June 1–30, 2022. The district is located 717 km from Addis Ababa and 150km east of Bahirdar, the capital city of the Amhara region. Its total population is 152,683 and according to 2022 data from the Andabet district administration office; the district has 26 kebeles with 34765 households. It has 1 primary health care center and 2 health posts. Its climate condition is Woinadega

### Study design

A community-based cross-sectional design was used with a systematic random sampling method June 1–30, 2022.

### Source and study population

The source and study population of the study were Mothers having children aged under nine years who had been living in the Andabet district, Northwest Ethiopia. Mothers who had at least one child of age less than nine years and those who lived in the district at least for six months were included in the study. On the contrary, mothers with mental illness, other serious systemic illnesses, and hearing problems were excluded from the study.

## Sample size determination

The sample size was calculated using a single population proportion formula for the proportion of poor TPP. By taking a similar study done in Oromia, Ethiopia with a proportion of 48.5% [1], 95% confidence level, 5% margin of error, 1.5 design effect, and 10% non-response rate, the final sample size for this study was determined to be 634.

## Sampling technique and procedure

A multistage sampling technique was used during the sampling process. Six Kebeles out of 26 Kebeles were selected by using a simple random sampling method after a list of kebeles was obtained from the Andabet district administration bureau. The total sample size of the study was allocated proportionally for each Kebele based on the number of mothers having children aged under nine years that were found in each Kebele. Finally, the households were chosen using a systematic random sample technique.

To carry out systematic random sampling, sampling frames were collected from each kebele. The total estimated number of the study population was 4565. Based on the study population and sample size required from each kebele, we calculated the interval between households and found it to be seven. Then the first household was randomly selected from 1 to 7 serial numbers of sampling and the remaining households were selected at every 7th interval. If mothers in the household did not fulfill the inclusion criteria and/or were not found with two repeated visits, the next household was taken (Fig 1).

## Variables of the study

**Dependent variables.**   The dependent variable was the mother's TPP.

**Independent variables.**   For this study, the independent variables were classified as individual and community-level variables. The individual level variables were the Age of the mother, Mother's education, Husband's education, Occupation of the mother, Religion,

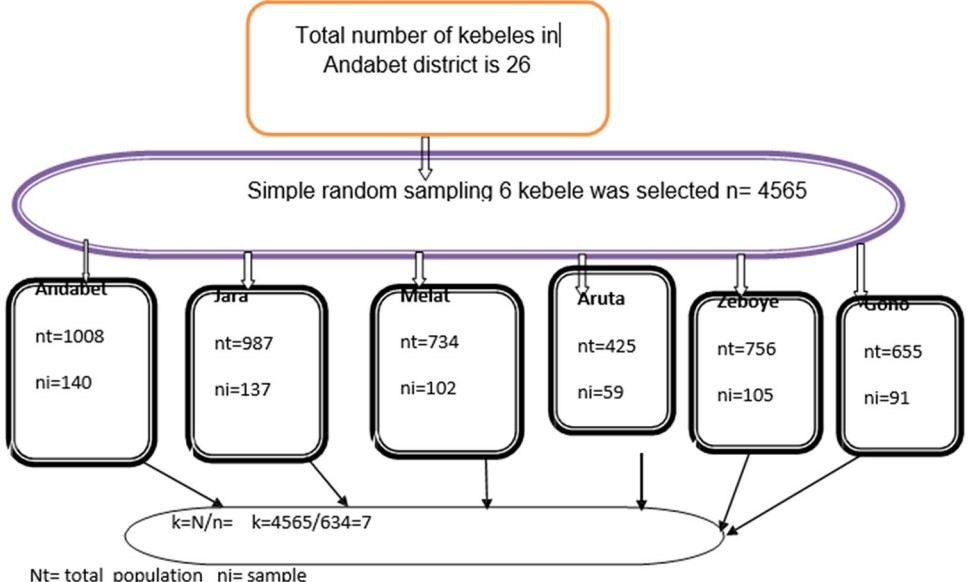

**Fig 1. Schematic presentation of sampling technique of trachoma prevention practice among mothers having children aged under nine years in Andabet district, Northwest Ethiopia, 2022:a multi-level analysis (n = 624).**

Marital status, Number of children under nine years, age of the youngest child, Sex of the youngest child, Health education, Time taken to the water point, Basic trachoma Knowledge, Attitude towards trachoma and Amount of water used per person per day. Community-level variables for this study were Residence, Type of water source, and Community women's illiteracy level.

## Operational definition

**Trachoma prevention practice assessment:** is the assessment of the mother's TPP towards F and E components of SAFE and is classified as good or poor based on the mean of the scores [1].

**Basic trachoma Knowledge assessment**: is the assessment of the mother's basic knowledge of trachoma and is classified as good or poor based on the mean of the scores [17].

**Attitude towards trachoma assessment:** this is the assessment of the mother's attitude towards trachoma and is classified as good or poor based on the mean of the scores [17,22,23].

**Facial cleanliness:** measured as an absence of ocular discharge, nasal discharge, and fly (ies) on the eye during the time of examination. If there was one from the list, considered not clean [24].

**Ocular discharge**: Any discharge around and/or in the eye at the time of examination [24].

**Nasal discharge**: any discharge seen in the nose at the time of examination [24].

**Fly-eye:** at least one fly contact with the eyelid margin during eye observation [7].

**Time taken to the water source**: collection time does not exceed 30 minutes.

**Availability of water:** An average person uses about 20 liters of water per day for domestic and personal hygiene [25].

**Utilization of waste disposal pit**: Disposal pits that had at least one of the following: discarded unwanted agricultural products, domestic products, or ashes (a burned sign of waste) were considered utilized, otherwise not [21].

**Latrine utilization:** Latrines that displayed at least two of the following during the observation: footpath to the latrine, fresh excreta inside the latrine, presence of a splash of urine, and the absence of a spider web of the squat were considered utilized, otherwise not [26].

**Cleanness of compound:** a household (residential) compound free from solid waste, liquid wastes, feces, animal dung, and domestic waste was considered clean [17].

**Health education**: Those who received education about trachoma and trachoma prevention at least once in the past two years were considered to have taken health education [21].

**Cleanness of latrine**: if there is at least one of these: human excreta out of the pit, stagnant urine, and unwanted trash on the floor of the latrine, it was not considered clean [27].

**Community-women illiteracy**: it is the aggregated community-level variable derived from maternal educational level and measured as the proportion of women with no formal education at the kebele/community level. Based on a median value it was then divided into low (mothers from communities with lower illiteracy levels) and high (mothers from communities with higher illiteracy levels) categories [28,29].

## Data collection procedure and quality control

Quantitative data was collected through a face-to-face interview (supported by observation when it is important) by using an interviewer-administered questionnaire, which was adapted from different literature and modified to the context. The questionnaire was first developed in English language and then translated into Amharic (the local language). The questionnaire has six different parts. Part-I: comprising of socio-demographic questions, Part-II: comprises fifteen different knowledge-assessing questions, Part-III: comprises seven different attitude-

assessing questions, Part-IV: comprises ten questions assessing the TPP, Part-V: comprises Environmental related questions and part-VI comprises the observation checklist.

Pretest was done on 5% of the total sample size at zeboye district in the south Gondar zone. After the pretest, necessary modifications and corrections took place to ensure validity. Four data collectors and one supervisor were recruited and trained for 1 day to collect and supervise the data respectively. The reliability of the question for TPP was checked by Cronbach alpha and the scale reliability coefficient was 0.795.

## Data processing and statistical analysis

Data was entered into Epi-Data version 4.6 then data cleaning, coding, and analysis were done using STATA version 16. Descriptive statistics were reported using text, tables, and figures. The proportion of poor TPP with its 95% Confidence interval (CI) was reported. A multilevel logistic regression analysis was used to assess factors associated with TPP to consider the hierarchical nature of the data in which mothers were nested within-cluster and mothers within the same cluster are more likely to share similar characteristics than mothers in another cluster which violates the independent assumptions of the standard logistic regression model such as the independent and equal variance assumptions.

While conducting a multilevel binary logistic regression analysis, we fitted both random effect and fixed effect analyses. The random effect parameter, intraclass correlation coefficient (ICC) quantifies the degree of heterogeneity of TPP between clusters and an ICC of more than 10% indicates that accounting for the cluster-level variability of TPP using multi-level analysis is appropriate. Moreover, proportion change in variance (PCV), and median odds ratio (MOR) were assessed.

In fixed effect analysis, four models were fitted; model 1 (with the outcome variable only), model 2 (incorporating individual-level variables), model 3 (fitted with community-level variables), and model 4 (incorporating both individual and community-level variables simultaneously). Among the four models fitted, the last model (model 4) was selected as the best-fitted model since it has the lowest deviance and highest PCV. For all models fitted, the adjusted odds ratio (AOR) with its 95% CI was reported. However, the interpretations are based on the final model, the best-fit model.

Both bivariable and multivariable multilevel logistic regression was done and variables with p-value <0.2 in the bivariable analysis were considered multivariable analysis. Finally, variables with p<0.05 in the multivariable multilevel analysis were declared to be significantly associated with TPP.

## Result

### Socio-demographic characteristics of study participants

A total of 624 study participants were included in the study. With a mean age of 2, the majority (79.49%) of mothers had children aged less than or equal to two years. Most, 88.78% of mothers didn't get health education about trachoma while more than half (57.2%) of them did not receive any formal education (Table 1).

### Environmental and related characteristics

Around three-fourths (73.24%) of mothers got water from the river and 342 (54.81%) of them travel more than 30 minutes to get water. Around two third (64.9%) and 354(56.73%) of mothers had poor basic trachoma knowledge and attitude towards trachoma. Regarding cleanness of the compound, 448 (71.79%) of mothers had unclean house compound (Table 2).

**Table 1.** Socio-demographic characteristics of mothers having children aged under nine years in Andabet, northwest Ethiopia, 2022: a multi-level analysis (n = 624).

| Variables | Category | Frequency | Percentage |
|---|---|---|---|
| Mother's age (in years) | 15–24 | 88 | 14.10 |
| | 25–34 | 312 | 50.00 |
| | 35yrs and above | 224 | 35.90 |
| Residence | Rural | 450 | 72.12 |
| | Urban | 174 | 27.88 |
| Religion | Orthodox | 575 | 92.15 |
| | Muslim | 49 | 7.85 |
| Marital status | Married | 463 | 74.2 |
| | Not currently married | 161 | 25.8 |
| Educational level Of mother | No formal education | 357 | 57.21 |
| | primary | 158 | 25.32 |
| | Secondary & above | 109 | 17.47 |
| Educational level Of father | No formal education | 233 | 50.32 |
| | primary | 117 | 25.27 |
| | Secondary & above | 113 | 24.41 |
| Occupation of mothers | farmer | 297 | 47.60 |
| | Housewife | 174 | 27.88 |
| | Government employee | 37 | 5.93 |
| | Merchant | 79 | 12.66 |
| | Daily laborer | 37 | 5.93 |
| Child age | Above 2 yrs. | 128 | 20.51 |
| | 2yrs and under 2yrs | 496 | 79.49 |
| Sex of the child | male | 321 | 51.44 |
| | female | 303 | 48.56 |
| Taking health Education about trachoma | yes | 70 | 11.22 |
| | No | 554 | 88.78 |
| Community-women Illiteracy level | high | 243 | 38.94 |
| | low | 381 | 61.06 |

## The magnitude of Trachoma prevention practice

In this study 49.84% (95%CI: 45.91%, 53.76%) of TPP was good and 50.16% (95%CI: 46.23%, 54.08%) TPP was poor (Fig 2). More than two third of mothers had a clean face, 427 (68.43%), and three fourth of children in the study had an unclean face, 457 (73.24%). More than three fourth (77.40%) of the mothers use latrines (Table 3).

## Random effect and model comparison

In the random effect analysis, in the null model, about 54% of the total variation in TPP occurred at the cluster level and is attributable to community-level factors. In addition, the null model also had the highest MOR value (6.51) indicating when randomly selecting a mother from one kebele with a higher risk of poor TPP and the other kebele at lower risk, mothers at the cluster (kebele) with a higher risk of poor TPP had 6.51 times higher odds of having a poor TPP as compared with their counterparts. Furthermore, the highest PCV (70.4%) in the final model (model 4) showed 70.4% of the variation in TPP across communities was explained by both individual and community-level factors. The model fitness was checked by using deviance and the model with the lowest deviance (model4) was the best-fitted model (Table 4).

**Table 2. -Environmental and other related characteristics of mothers having children aged under nine years in Andabet district, northwest Ethiopia, 2022: a multilevel analysis (n = 624).**

| Variables | Category | Frequency | Percentage |
|---|---|---|---|
| Source of water | River | 457 | 73.24 |
| | Household tap | 167 | 26.76 |
| Time taken to Water point | < = 30 min | 282 | 45.19 |
| | >30 min | 342 | 54.81 |
| Amount of water used per Person per day | < = 20 liter | 128 | 20.51 |
| | >20 liter | 496 | 79.49 |
| Frequency of Getting water | All the time | 516 | 82.69 |
| | Either day or night | 17 | 2.72 |
| | In more than a day | 91 | 14.59 |
| Source of energy for cooking | | | |
| electricity | Yes | 12 | 1.92 |
| | No | 612 | 98.08 |
| wood | Yes | 615 | 98.56 |
| | No | 9 | 1.44 |
| Animal dung | Yes | 610 | 97.76 |
| | No | 14 | 2.24 |
| Charcoal | Yes | 278 | 44.62 |
| | No | 345 | 55.38 |
| Type of household latrine used | Covered pit latrine | 414 | 84.49 |
| | Uncovered latrine | 76 | 15.51 |
| Cleanness of latrine | Clean | 327 | 67.01 |
| | Not clean | 161 | 32.99 |
| Availability of hand washing Material near to latrine | Yes | 15 | 3.07 |
| | No | 473 | 96.93 |
| Cleanness of the home compound | Clean | 176 | 28.21 |
| | Not clean | 448 | 71.79 |
| Availability of community latrine | Yes | 153 | 24.52 |
| | No | 471 | 75.48 |
| Basic trachoma Knowledge | Good | 219 | 35.1 |
| | Poor | 405 | 64.9 |
| Attitude towards Trachoma | Good | 354 | 56.73 |
| | Poor | 270 | 43.27 |

## Factors associated with trachoma prevention practice

In the multivariable multi-level logistic regression, maternal education level, maternal occupation, time taken to the water point, and health education about trachoma were significantly associated with poor TPP.

Mothers with no formal education had 2.95(AOR = 2.95; 95%CI: 1.41, 6.15) times higher odds of poor TPP as compared to those mothers with secondary education or above. Mothers with primary education had 2.33 (AOR = 2.33; 95%CI: 1.04, 5.24) times higher odds of poor TPP as compared to those mothers with secondary education or above. On maternal occupation, mothers who were farmers had 3.02(AOR = 3.02; 95%CI: 1.73, 5.28) times higher odds of poor TPP as compared to those mothers who were housewives. Mothers who were merchants had 2.63 (AOR = 2.63; 95%CI: 1.20, 5.75) times higher odds of poor TPP as compared to those mothers who were housewives. Regarding time taken to the water point, mothers who traveled >30 minutes to the water point had 4.60 (AOR = 4.60, 95CI:1.30, 16.26) times higher odds of

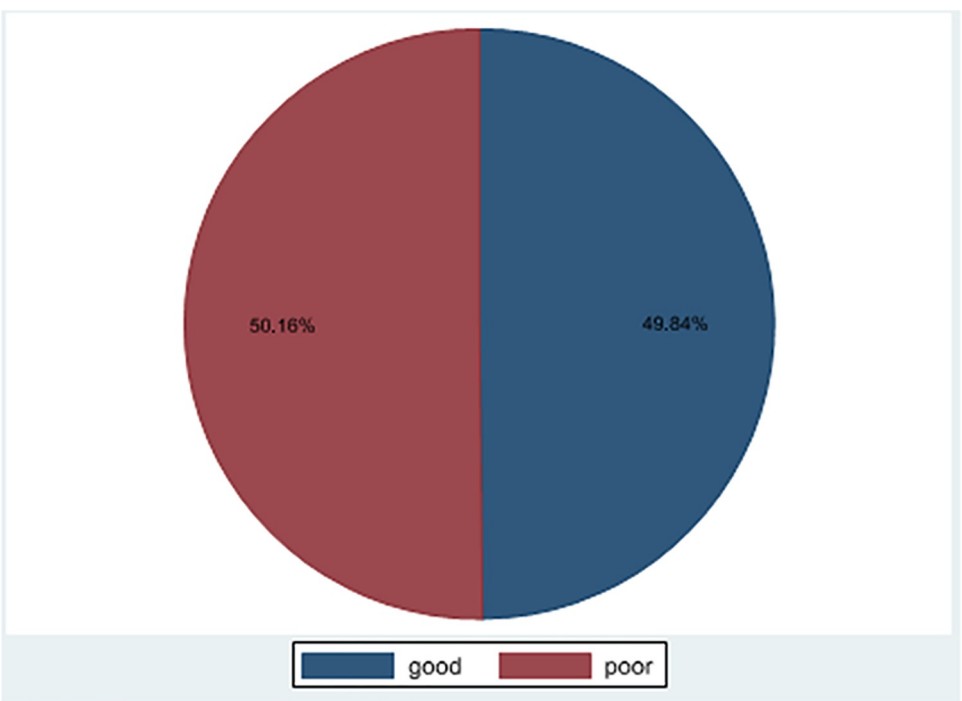

**Fig 2. Trachoma prevention practice among mothers having children aged under nine years in Andabet district, Northwest Ethiopia,2022:a multi-level analysis (n = 624).**

**Table 3.** -Trachoma prevention practice and associated factors among mothers having children aged under nine years in Andabet district, northwest Ethiopia,2022: a multi-level analysis (n = 624).

| Variables | Category | Frequency | Percentage |
|---|---|---|---|
| Mother facial cleanness | Clean | 427 | 68.43 |
| | Not clean | 197 | 31.57 |
| Child facial cleanness | Clean | 167 | 26.76 |
| | Not clean | 457 | 73.24 |
| Using soap for face washing | Yes | 491 | 78.69 |
| | No | 133 | 21.31 |
| Did not share fomites With family | Yes | 460 | 73.72 |
| | No | 164 | 26.28 |
| Separated house for Animal dwelling | Yes | 296 | 47.44 |
| | No | 328 | 52.56 |
| Availability of household Latrine | Yes | 488 | 78.21 |
| | No | 136 | 21.79 |
| Infant feces disposal To latrine | Yes | 482 | 77.24 |
| | No | 142 | 22.76 |
| Utilization of latrine | Yes | 483 | 77.40 |
| | No | 141 | 22.60 |
| Availability of waste Disposal pit | Yes | 289 | 46.39 |
| | No | 334 | 53.61 |
| Utilization of waste Disposal pit | Yes | 293 | 46.96 |
| | No | 331 | 53.04 |

**Table 4.** -random effect analysis in trachoma prevention practice among mothers having children aged under nine years in Andabet, Northwest Ethiopia,2022 (n = 624).

| Parameter | Model 1 | Model 2 | Model 3 | Model 4 |
|---|---|---|---|---|
| MOR | 6.51 | 2.90 | 3.77 | 2.67 |
| PCV | Reff. | 0.678 | 0.497 | 0.706 |
| ICC | 0.54 | 0.27 | 0.37 | 0.25 |
| Deviance | 583.69 | 536.13 | 576.34 | 531.74 |

poor TPP as compared to those mothers who traveled ≤30 minutes. Mothers who didn't receive health education about trachoma had 2.36 (AOR = 2.36; 95%CI: 1.16, 4.79) times higher odds of poor TPP compared to their counterparts (Table 5).

**Table 5.** -Multilevel logistic regression analysis factors associated with trachoma prevention practice among mothers having children aged under nine years in Andabet district, Northwest Ethiopia,2022:a multi-level analysis (n = 624).

| variables | Model 1 | Mode 2 AOR 95%(CI) | Mode 3 AOR 95%(CI | Mode 4 AOR 95%(CI) |
|---|---|---|---|---|
| Maternal education | | | | |
| No formal education | | 2.97(1.42,6.19) | | 2.95(1.41,6.15)** |
| Primary | | 2.36 (1.05, 5.32) | | 2.33(1.04,5.24)* |
| Secondary & above | | 1.00 | | 1.00 |
| Maternal occupation | | | | |
| housewife | | 1.00 | | 1.00 |
| farmer | | 3.16(1.82,5.50) | | 3.02(1.73,5.28)*** |
| Government employee | | 1.40(0.45,4.35) | | 1.39(0.44,4.37) |
| Merchant | | 2.64(1.21,5.75) | | 2.63(1.20,5.75) * |
| Daily laborer | | 1.64(0.61,4.41) | | 1.61(0.60,4.36) |
| Frequency of getting water | | | | |
| All the time | | 1.00 | | 1.00 |
| Day or night | | 0.45(0.10,2.32) | | 0.51(0.10,2.50) |
| In > a day | | 0.48(0.2,1.13) | | 0.50(0.21,1.19) |
| Time taken to water point | | | | |
| ≤30 minute | | 1.00 | | 1.00 |
| > 30 minutes | | 5.85(1.70,20.10) | | 4.60(1.30,16.26)* |
| Health education about trachoma | | | | |
| No | | 2.25(1.11,4.57) | | 2.36(1.16,4.79)* |
| Yes | | 1.00 | | 1.00 |
| Source of water | | | | |
| River | | | 0.45(0.07,3.15) | 0.54(0.07,4.26) |
| Household tap | | | 1.00 | 1.00 |
| Residence | | | | |
| Rural | | | 3.53(0.53,23.54) | 3.16(0.42,23.70) |
| Urban | | | 1.00 | 1.00 |
| Community-women illiteracy | | | | |
| Low | | | 0.09(0.01,0.87) | 0.46(0.1,3.55) |
| High | | | 1.00 | 1.00 |

Note

*** = P<0.001

** P<0.01, and

* = P<0.05

## Discussion

The study aimed to assess the magnitude and associated factors of TPP in the andabet district, Northwest Ethiopia. According to the finding of this study, the magnitude of poor TPP was 50.16%. This finding is in line with a study conducted in Oromia Ethiopia [1]. However, this magnitude of poor TPP was found lower compared to a study conducted in Tigray [17] and higher than a study conducted in north Vietnam and the Lemo district of Southern Ethiopia [21,30]. The discrepancy might be due to the difference in the study population as in most of the indicated studies (except the study in Oromia, Ethiopia, 2021) children under nine years were their study subjects. Besides, most of the above studies were based on smaller sample size. The other possible explanation might be the study period and the difference in the availability and accessibility of maternal health services and facilities. Moreover, the discrepancy of this finding with that of the findings of studies conducted out of Ethiopia might be due to socio-demographic and cultural differences.

The study at hand found that Mothers with no formal education and mothers with primary education are more likely to have poor prevention practice as compared with those mothers with secondary or above. This is supported by a study done in Vietnam [30], which similarly showed that those with no formal education are more likely to have poor prevention practice. This might be due to the levels and ways of understanding regarding the mechanism of transmission, prevention measures, and negative effects of the diseases being different among mothers with different levels of education. That is educated mothers would likely appreciate the problems related to poor prevention practice more than those with no formal education [30,31].

In this study, health education is another important variable significantly associated with TPP. That is mothers who didn't receive health education about trachoma were more likely to have poor TPP as compared to their counterparts. This finding is supported by a study conducted in the lemo district [21]. Such a correlation could be because mothers who have not attended health education programs lack the skills needed to prevent trachoma. Hence, they are more likely to have cultural misconceptions about how to use water for environmental sanitation and personal hygiene [32,33].

Moreover, in this study mothers who were farmers and merchants were more likely to have poor TPP as compared to those mothers who were housewives. It might be because based on our study 88% of farmers and 93.6% of merchants did not receive health education programs on trachoma. In addition, most (81.5%) of them have to travel more than 30 minutes to get water making it more difficult for them to clean their face and improve their environment than those who are housewives.

Consistent with other studies conducted in Kenya and Oromia [1,34], in this study, the time taken to the water point is significantly associated with TPP. That is mothers who traveled more than 30 minutes to the water point had higher odds of poor TPP as compared to their counterparts. This might be because access to and adequacy of water differs between mothers who travel over 30 minutes and those who travel less than 30 minutes. Mothers who have insufficient water may not be able to use it for facial and environmental cleanliness. Furthermore, based on our study, the majority (81.5%) of mothers traveling over 30 minutes for water get it from unclean streams. while most (74%) of those mothers traveling less than 30 minutes get it from relatively clean personal and public pipes [34].

### Strengths and limitations of the study

There were strengths and shortcomings in this study. To begin with the strength, this study explored neglected tropical disease that became hyper-endemic in our study area after the

implementation of SAFE for about 8 to 11 years. So the result of the study would be important for employing combined efforts to address identified modifiable risk factors and will have significant policy implications in providing support to the affected community. Besides, the study uses multi-level modeling taking into account the clustering effect to draw valid inference and conclusion. Moreover, to ensure representativeness, the study uses an adequate sample size. However, this study had limitations as it's a cross-sectional study. It may not show a true temporal relationship between the outcome and the independent variables. Besides, due to the lack of sign language-trained data collectors and the inability to obtain psychiatric therapists as data collectors, it was not possible to include those mothers with mental illness and hearing problems, although we do not expect a significant number of such women in the targeted communities. Moreover, the study mainly relies on the mother's self-report, so there may be a chance of recall bias. Furthermore, Social desirability bias might be introduced while assessing sensitive variables.

## Conclusion

In this study, the magnitude of poor TPP was high relative to other studies. Those mothers with no formal education, with primary education, those who take more time to water point (>30 minutes), those who didn't receive health education about trachoma, and those who were farmers and merchants were at higher odds of poor TPP. Therefore, special attention should be given to these high-risk groups so that this devastating health problem can be decreased.

## Recommendations

To governmental and non-governmental organizations: focus on facial cleanliness (F) and Environmental improvement (E) components of the WHO recommended SAFE strategy for the elimination of trachoma especially in highly endemic countries like Ethiopia.

To Amhara region trachoma control program: planning health education program and enhancing water supply are recommended to improve TPP overall.

To the health office in Andabet district: special attention should be given to those mothers with no formal education and primary education and who reside far from a water source. Besides, health education programs about trachoma should be strengthened.

To the global trachoma community: The low coverage in "Trachoma Prevention Practice" will negatively affect the efforts of the WHO-launched global alliance for the elimination of blinding Trachoma. Therefore, it is recommended to give special attention to those communities regarding the Facial cleanliness (F) and Environmental Improvement (E) components of the SAFE strategy.

## Supporting information

**S1 Dataset. Data TPP.**
(ZIP)

## Acknowledgments

We are grateful to the Department of Optometry, study participants, and data collectors.

## Author Contributions

**Conceptualization:** Zufan Alamrie Asmare.

**Data curation:** Zufan Alamrie Asmare.

**Formal analysis:** Zufan Alamrie Asmare.

**Funding acquisition:** Zufan Alamrie Asmare.

**Investigation:** Zufan Alamrie Asmare.

**Methodology:** Zufan Alamrie Asmare.

**Resources:** Zufan Alamrie Asmare.

**Software:** Zufan Alamrie Asmare.

**Supervision:** Zufan Alamrie Asmare.

**Validation:** Zufan Alamrie Asmare.

**Visualization:** Zufan Alamrie Asmare, Natnael Lakachew Assefa, Dagmawi Abebe, Solomon Gedlu Nigatu, Yezinash Addis Alimaw.

**Writing – original draft:** Zufan Alamrie Asmare.

**Writing – review & editing:** Zufan Alamrie Asmare, Natnael Lakachew Assefa, Dagmawi Abebe, Solomon Gedlu Nigatu, Yezinash Addis Alimaw.

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
