## [Decision Letter · Decision Letter 0]

6 Mar 2023

Dear Miss Asmare,

Thank you very much for submitting your manuscript "Trachoma prevention practice and associated factors among mothers having children aged under nine years in Andabet district, northwest Ethiopia, 2022 :a multi-level analysis" for consideration at PLOS Neglected Tropical Diseases. As with all papers reviewed by the journal, your manuscript was reviewed by members of the editorial board and by several independent reviewers. In light of the reviews (below this email), we would like to invite the resubmission of a significantly-revised version that takes into account the reviewers' comments. 

We cannot make any decision about publication until we have seen the revised manuscript and your response to the reviewers' comments. Your revised manuscript is also likely to be sent to reviewers for further evaluation.

Sincerely,

Joseph M. Vinetz

Section Editor

Joseph Vinetz

Section Editor

Reviewer's Responses to Questions

**Key Review Criteria Required for Acceptance?**

**Methods**

-Are the objectives of the study clearly articulated with a clear testable hypothesis stated?

-Is the study design appropriate to address the stated objectives?

-Is the population clearly described and appropriate for the hypothesis being tested?

-Is the sample size sufficient to ensure adequate power to address the hypothesis being tested?

-Were correct statistical analysis used to support conclusions?

-Are there concerns about ethical or regulatory requirements being met?

Reviewer #1: The study is very important for the global trachoma community for decision making at global/international, national, regional, district and community levels by all stakeholders involved as Ethiopia currently has an estimated 50% of the global population requiring interventions for trachoma elimination. Addressing the trachoma problem in Ethiopia is potentially wiping out half the global trachoma problem. There is enough information that shows that the study is important and worthy of publication. However, major revision is needed. The overall presentation is poor. Grammar is poor. I suggest the authors find someone who can proofread and the many grammatic mistakes.

Objectives should be revised. See my comments in the attached file. They are not clearly articulated.

I believe that the study design is appropriate to address the stated objectives.

The population is clearly described and appropriate for the hypothesis being tested.

The sample size looks sufficient to ensure adequate power to address the hypothesis being tested.

I believe that correct statistical analysis was used to support conclusions although I may not ne strong enough on statistical methods.

Selection of participants is generally ethical and meets regulatory requirements. However, some women were excluded with weak justification for the exclusion. Better explanation needed in the limitation section.

**Results**

-Does the analysis presented match the analysis plan?

-Are the results clearly and completely presented?

-Are the figures (Tables, Images) of sufficient quality for clarity?

Reviewer #1: The study is a replication of similar studies conducted in other parts of the country and in other countries. The analysis matches the analysis plan. Results are completely presented. The authors need to improve the presentation in general.

**Conclusions**

-Are the conclusions supported by the data presented?

-Are the limitations of analysis clearly described?

-Do the authors discuss how these data can be helpful to advance our understanding of the topic under study?

-Is public health relevance addressed?

Reviewer #1: Conclusions are supported by the data presented but presentation is generally poor.

The limitations of analysis are described but more work needed.

The authors tried to discuss how these data can be helpful to advance our understanding of the topic under study, but more work is needed to improve the presentation of the study.

Authors have poorly addressed the public health relevance of the study. I have made some suggestions in the attached file.

**Editorial and Data Presentation Modifications?**

Reviewer #1: I have added my editorial suggestions in the attached file.

**Summary and General Comments**

Reviewer #1: The study is a replication of studies conducted in other countries and in other parts of Ethiopia to assess community practices that influence trachoma prevention and elimination. Half the global population requiring intervention for trachoma elimination is in Ethiopia where some regions/districts have up to 37% TF rate (hyperendemicity) after years of A treatment. This study demonstrates the need to consider support for the introduction of other interventions (F and E) for trachoma elimination in Ethiopia and thus elimination of an estimated half the global burden. I believe that the essential information needed (objectives, methodology, result) is available in this draft of the manuscript. However, overall presentation is poor with too many grammatic errors. Authors should take time to improve the presentation including finding someone who can proofread and edit the manuscript for them. I have made some suggestions in the attached file.

PLOS authors have the option to publish the peer review history of their article (what does this mean?). If published, this will include your full peer review and any attached files.

Reviewer #1: No
---

## [Decision Letter · Decision Letter 1]

11 May 2023

Dear Miss Asmare,

Thank you very much for submitting your manuscript "Trachoma prevention practice and associated factors among mothers having children aged under nine years in Andabet district, northwest Ethiopia, 2022: A multi-level analysis" for consideration at PLOS Neglected Tropical Diseases. As with all papers reviewed by the journal, your manuscript was reviewed by members of the editorial board and by several independent reviewers. The reviewers appreciated the attention to an important topic. Based on the reviews, we are likely to accept this manuscript for publication, providing that you modify the manuscript according to the review recommendations. 

Sincerely,

Joseph M. Vinetz

Section Editor

Joseph Vinetz

Section Editor

Reviewer's Responses to Questions

**Key Review Criteria Required for Acceptance?**

**Methods**

-Are the objectives of the study clearly articulated with a clear testable hypothesis stated?

-Is the study design appropriate to address the stated objectives?

-Is the population clearly described and appropriate for the hypothesis being tested?

-Is the sample size sufficient to ensure adequate power to address the hypothesis being tested?

-Were correct statistical analysis used to support conclusions?

-Are there concerns about ethical or regulatory requirements being met?

Reviewer #1: Manuscript was revised well needing only minor revisions before publication.

**Results**

-Does the analysis presented match the analysis plan?

-Are the results clearly and completely presented?

-Are the figures (Tables, Images) of sufficient quality for clarity?

Reviewer #1: Manuscript was revised well needing only minor revisions before publication.

**Conclusions**

-Are the conclusions supported by the data presented?

-Are the limitations of analysis clearly described?

-Do the authors discuss how these data can be helpful to advance our understanding of the topic under study?

-Is public health relevance addressed?

Reviewer #1: Manuscript was revised well needing only minor revisions before publication.

**Editorial and Data Presentation Modifications?**

Reviewer #1: Most of the revision needed is on use of acronym. In line 28 please write "trachoma prevention practices" in full followed by the acronym in bracket (TPP). Afterwards, just put TPP. Use the acronym TPP instead of in full in line 31, 47,49, 86-87, 91,92,93, 94,96,118, 151, 160-161, 200,205,210,211-212, 218, 219-220, 231, 252,253, 264-265, 267, 268-269, 269, 271, 281, 283, 283, 287, 289, 291-292, 294, 301, 303,304, 321-322, 323, 329, 335, 336-337, 359, 362, 369. I might have missed some. Please go through and change accordingly. 

Abstract: World Health Organization (WHO). I suggest you use 'cleanliness' throughout when referring to the SAFE strategy. Put TPP in bracket after trachoma prevention practices as you have used the acronym later in the abstract. On dates, I believe you say from June 5 to June 10, 2022. Remove 'from' and the date is still good (was conducted June 1-30, 2022. 

Line 25: World Health Organization (WHO).

Line 34: remove 'from'

Line 37: small 'v'

Line 57: for consistency, please use 'cleanliness'.

Line 74: remove the 's' from disease, use singular as it is just trachoma.

Line 84-85: please improve on this sentence as it is vague, not clear enough what you are saying.

Line 87: small 's'.

Line 100 and 107: remove 'from'.

Line 119: no brackets please.

Line 120: add comma after 'rate'.

Line 127: please replace 'of' with 'that'.

Line 129: small 'c'.

Line 143: please replace highlighted words with 'were' and add a comma after 'visits'.

Line 155: comma after status.

Line 164: add a full stop after the bracket.

Line 167: 'ies' in the bracket, not just 's' as plural of fly is flies.

Line 174: small 'd'.

Line 189: dived or divided?

Line 195: use comma, then small 'w'.

Line 235: capital 'M'.

Line 246: full stop after bracket.

Line 286: Capital 'O'.

Line 306: capital 'S'.

Line 307: i suggest you use 'as in' or because in'.

Line 308: comma after 'besides'.

Line 309: I suggest you use 'smaller'.

Line 309: better to use 'might be'.

Line 310: 'difference' better.

Line 315: use comma, not full stop.

Line 322: better to use 'receive".

Line 328: comma after moreover.

Line 331: space, then capital 'I'.

Line 337: suggest you use 'because'.

Line 338: full stop after minutes.

Line 345: 'a neglected tropical disease', not diseases as you are referring to just trachoma.

Line 346: full stop please.

Line 347: in place of 'to', use 'to address'.

Line 348: Full stop after 'community'. Then start next sentence with a capital B.

Line 351: full stop instead of comma.

Line 352: small 'd'.

Line 354-355: sounds negative as written here. I suggest instead "although we do not expect a significant number of such women in the targeted communities".

Line 362-363-364: please rewrite. I suggest " Therefore, special attention should be given to these high-risk groups so that this devastating health problem can be decreased".

Line 366: cleanliness.

Line 367: I suggest 'the WHO-recommended SAFE strategy for elimination of trachoma especially in highly endemic countries like Ethiopia".

Line 371: use 'reside far".

Line 373-374: I suggest "will negatively affect efforts of the ". 

Line 374: elimination, not illumination.

**Summary and General Comments**

Reviewer #1: I believe that this is a good study conducted in Ethiopia and that this manuscript will again highlight the need for special support for the implementation of the F and E components of the WHO-recommended SAFE strategy in countries like Ethiopia that currently has about 50% of the global trachoma burden if the disease is to be eliminated as a public health problem globally. The authors have significantly improved the presentation.

PLOS authors have the option to publish the peer review history of their article (what does this mean?). If published, this will include your full peer review and any attached files.

Reviewer #1: No

Figure Files:

Data Requirements:

Reproducibility:

References

---

## [Editor Report · Decision Letter 2]

2 Jun 2023

Dear Miss Asmare,

We are pleased to inform you that your manuscript 'Trachoma prevention practice and associated factors among mothers having children aged under nine years in Andabet district, northwest Ethiopia, 2022: A multi-level analysis' has been provisionally accepted for publication in PLOS Neglected Tropical Diseases.

Best regards,

Joseph M. Vinetz

Section Editor

Joseph Vinetz

Section Editor

---

## [Editor Report · Acceptance letter]

16 Jun 2023

Dear Miss Asmare,

We are delighted to inform you that your manuscript, "Trachoma prevention practice and associated factors among mothers having children aged under nine years in Andabet district, northwest Ethiopia, 2022: A multi-level analysis," has been formally accepted for publication in PLOS Neglected Tropical Diseases.

Best regards,

Shaden Kamhawi

co-Editor-in-Chief

Paul Brindley

co-Editor-in-Chief
